# Optimizing the Release Pattern of *Telenomus podisi* for Effective Biological Control of *Euschistus heros* in Soybean

**DOI:** 10.3390/insects15030192

**Published:** 2024-03-14

**Authors:** William Wyatt Hoback, Gabryele Ramos, Rafael Hayashida, Daniel Mariano Santos, Daniel de Lima Alvarez, Regiane Cristina de Oliveira

**Affiliations:** 1Department of Entomology and Plant Pathology, Oklahoma State University, Stillwater, OK 74078, USA; rafael.hayashida@okstate.edu; 2Crop Protection Department, School of Agronomic Sciences, São Paulo State University “Júlio de Mesquita Filho” (FCA/UNESP), Botucatu 18610034, SP, Brazil; gabryele.sr@gmail.com (G.R.); d_m.s@hotmail.com (D.M.S.); daniel-alvarez92@hotmail.com (D.d.L.A.); regiane.cristina-oliveira@unesp.br (R.C.d.O.)

**Keywords:** biological control, augmentative release, dispersal, Pentatomidae, *Glycine max* L.

## Abstract

**Simple Summary:**

The brown stink bug is a serious pest of soybeans in South America, and its control is challenging because of its resistance to chemical insecticides. As a result, the egg parasitoid *Telenomus podisi* has been tested in laboratory and field conditions as a biological control agent that attacks the brown stink bug eggs. The parasitoid is released into fields as adults or parasitized eggs and the effectiveness depends on its ability to find hosts. In this work, we evaluated the dispersal of *T. podisi* and determined a dispersal capacity influenced by soybean growth stage that varied between 31 and 39 m. The maximum parasitism rate of stink bug eggs was about 60%. Based on these results, we recommend that *T. podisi* release points are spaced at a maximum of 30 m apart in order to provide sufficient control of pest stink bugs.

**Abstract:**

An augmentative biological control program using the egg parasitoid *Telenomus podisi* Ashmead (Hymenoptera: Platygastridae) is a promising tool for the management of the brown stink bug, *Euschistus heros* (Fabricius) (Hemiptera: Pentatomidae) in soybeans. The *T. podisi* are released as adults or pupae within lab-reared *E. heros* eggs. Because of the small size of the parasitoid and potentially limited dispersal ability, determining the optimal release pattern is critical for biological control of the target pest. This study used sentinel *E.* heros eggs to investigate *T. podisi* dispersal within soybean crops during two distinct phenological stages: the beginning of flowering (Vn–R1) and the grain-filling phase (R5–R6). Data were analyzed using semi-variograms and kriging maps. The results indicate significant differences in parasitism rates between the two plant growth stages and among different matrices. The maximum dispersal range for *T. podisi* was calculated at 39.0 m in the Vn–R1 stage with a maximum parasitism rate of 42%, while in the R5–R6 stage, the maximum dispersal range was calculated to be 30.9 m with a maximum parasitism of 73%. Therefore, it is recommended that release points for *T. podisi* be spaced no further than 30 m apart. These results provide valuable insights for future research and applications in biological control strategies, including adjustment of the logistics and release technique depending upon the crop phenological stage.

## 1. Introduction

The adoption of egg parasitoids in biological control has proven to be highly effective in managing various pests, particularly defoliating caterpillars (Lepidoptera: Noctuidae and Erebidae) and stink bugs (Hemiptera: Pentatomidae) that pose threats to soybean crops around the world [1,2]. One of the most-studied egg parasitoids, *Telenomus podisi* Ashmead, 1893 (Hymenoptera: Scelionidae), is recognized as an effective natural enemy for the key stink bug pests that attack soybean crops [3]. Following the commercial release of *T. podisi* in 2019 in Brazil [4], more robust programs and studies have been conducted to improve the use of this biological control agent in combatting stink bug pests of soybeans [5,6,7,8].

To assess the effectiveness of an egg parasitoid, a number of biological characteristics must be considered, especially when developing management strategies and field release methodologies. Among the most critical parameters are the development rate, longevity, fecundity, sex ratio, and dispersal capacity of the parasitoid [9]. While numerous studies have been conducted on laboratory mass rearing of *T. podisi* [9,10,11], and the biotic [12] and abiotic [6] factors influencing its life history, research on *T. podisi*’s dispersal ability in soybean remains poorly studied. Thus, studies are required to assess the dispersal ability of *T. podisi* and to assess its host-finding ability in plants at different phenological stages. This reassessment should occur in an area after the introduction of *T. podisi*, where its natural occurrence is simultaneously quantified and where additional nearby biological releases do not occur [13].

Information on the dispersal ability of *T. podisi* allows calculation of the number of release points to ensure a more homogeneous distribution of parasitism and, consequently, whole field control of stink bug populations. Moreover, assessing whether dispersal changes during different soybean phenological stages is essential for developing comprehensive biological control programs for stink bug species. Thus, this study used semi-variograms and kriging maps to assess the dispersal of *T. podisi* to infest sentinel *E. heros* eggs within a soybean crop during two distinct soybean phenological stages, at the beginning of flowering and the grain-filling phase. This knowledge can allow adjustments to the release strategy, whether it involves individual capsules or bulk pupae delivered using unmanned aerial vehicles (UAVs or drones) at selected release points [14].

## 2. Material and Methods

### 2.1. Laboratory Rearing of E. heros and T. podisi

The rearing of *E. heros* was conducted at the Department of Plant Protection at the Experimental Farm Lageado, Botucatu Campus, São Paulo (latitude 22°53′09″ S, longitude 48°26′42″ W). Insects were collected from soybean fields at an experimental farm (geographic coordinates latitude 22°48′19.5″ S, longitude 48°25′38.4″ W). They were reared in a laboratory under controlled conditions of temperature (25 ± 2 °C), relative humidity (70 ± 5%), and a photoperiod of 14 h light to 10 h darkness, following procedures similar to those described by Silva et al. [11] and Borges et al. [15]. Nymphs from the third instar and adults of *E. heros* were kept in transparent cages (4.5 L plastic pots). The lids for the cages had an opening covered with organza fabric to allow ventilation of the cage and the bottom of the cages were covered with filter paper. Insects were fed a natural diet consisting of bean pod seeds (*Phaseolus vulgaris*, Linnaeus), raw peanuts (*Arachis hypogaea*, Linnaeus), and seasonally with privet fruits (*Ligustrum lucidum*, Linnaeus). Additionally, to ensure adequate moisture, a plastic plate (60 mm) containing cotton soaked in distilled water was placed into each cage.

When *E. heros* reached the adult stage, four pieces of raw cotton fabric measuring 10 × 5 cm were placed on the cage walls as a substrate for oviposition. The density of adults inside the cage was approximately 100 male–female pairs and egg masses were collected every two days. The removed egg masses were placed in plastic capsules with a diameter of 60 mm, with a 5 cm fragment of green bean pod, until the eggs hatched. Once they completed the second instar, the insects were transferred to cages and fed as above.

The multiplication of *T. podisi* was carried out using *E. heros* eggs and were reared following methods described by Peres and Corrêa-Ferreira [16]. The *E. heros* eggs were glued to paper cards (8 × 10 cm), introduced into plastic cages (height 8.5 cm, diameter 7 cm), and offered to adult *T. podisi* for oviposition. Small droplets of honey produced by *Apis mellifera* L. were placed inside these cages to feed adults. The cages were subsequently sealed, and parasitism was permitted for 24 h. Following this, the eggs were extracted and placed in new pots, sealed once again with plastic film, and maintained under the same laboratory conditions. The adults from these eggs were then utilized for experiments or colony maintenance.

### 2.2. Evaluation of Dispersal in the Field

The dispersal tests were performed on 6 January and 18 February 2020, in a commercial field situated in Santa Cruz do Rio Pardo, São Paulo (latitude 22°49′51.46″ S, longitude 49°19′16.35″ W) with recommended commercial preparation for a soybean farm. However, no insecticides or fungicides were used in the field during the trials.

The evaluation of *T. podisi* dispersal in the field was conducted with a georeferenced demarcation of matrices, covering a total area of 6400 m^2^. The experiment was arranged in a completely randomized design with six replications, where each matrix constituted one replication. Every matrix consisted of 63 equidistant points (13.33 m east-west and 10 m north-south). In addition, a seventh plot was established, but without the release of *T. podisi* and served as a control to assess natural rates of parasitism. The soybean cultivar used was Nidera 6700 IPRO, which is characterized as a later-maturing cultivar (maturity group 6.7) with an indeterminate growth habit [17].

At each point of the matrix, a 1.50 m red fiberglass pole was positioned on which a white tulle fabric card containing 30 sentinel eggs of *E. heros* was fixed with wire (Figure 1). In order to ensure that all the *E. heros* embryos were dead, the sentinel eggs were cryopreserved in liquid nitrogen right before being used in the test. At the central pole of each demarcated test matrix, about 6500 newly emerged *T. podisi* females were released following the recommended density specified in the macrobiological control agent’s reference documentation [4] and provided with drops of pure honey ad libitum.

Each card containing the sentinel eggs was collected 24 h after the females were released. The cards were then kept in a climate-controlled chamber at 25 °C, 60% RH, and 14 h photophase. They were evaluated separately until the emergence of *T. podisi* adults. The values obtained in the control area were used to correct the parasitism obtained in the *T. podisi* release matrices.

The test was carried out during the Vn–R1 stage (end of the vegetative period—beginning of flowering) and repeated during the R5–R6 stage (beginning to end of grain filling) [18] using the same points and matrices. Temperature (T °C), relative humidity (%), wind speed (m∙s^−1^) and direction were collected when parasitoids were released at the center of each matrix using a digital anemometer (model Tan 100 Incoterm T-ANE-0010, Contagem, MG, Brazil). Weather data were also obtained for the 24 h release periods from the nearest weather station at Agroterenas S/A.

Percent parasitism was determined for each point of the six release sites. The maximum dispersal distance was determined based on the geostatistical analyses of the parasitism obtained at each point. For further analysis, we grouped the poles into three boxes around the center of release: Inner, middle, and outer boxes (Figure 1). The inner box included the poles around the center of release and the central pole (consisting of 9 poles). The middle box included the subsequent poles around the inner (consisting of 16 poles), and the outer box included the remaining poles (consisting of 38 poles).

### 2.3. Statistical Analyses

The average parasitism rates among matrices were compared using an ANOVA test, and the averages between the soybean stages (Vn–R1 and R5–R6) were compared using a Student’s *t*-test. The statistical analyses were conducted in the R computing environment, utilizing the ‘AgroR’ package [19] and “ggplot2” package [20]. Before proceeding with the analysis, we performed exploratory data analysis to assess the assumptions of normality of residuals [16] and homogeneity of variances [21].

The average distance of dispersal and the areas reached by *T. podisi* were inferred from the percentage of parasitism of sentinel eggs arranged in matrices of equidistant points (georeferenced in UTM units). Patterns were verified through the generation of semi-variograms and kriging maps using GS+ geostatistics 9.00 software (Gamma Design Software 9.0 version).

## 3. Results

The release of parasitoids increased the rate of sentinel stink bug egg parasitism for all replicates (Figure 2A). The natural parasitism of *E. heros* eggs in the control areas was approximately 0.30% in the Vn–R1 stage and 0.32% in the R5–R6 soybean stage, indicating a low natural incidence of *T. podisi* in the study area (Figure 2B). The release of parasitoids increased the rate of parasitism and resulted in about 2.66 ± 0.63% infection rate of all eggs with release in the vegetative period and about 5.03 ± 0.98% in the pod stage (Figure 2A).

During the phenological stage Vn–R1, matrix M2 had the highest average parasitism rate (4.32 ± 0.82%), while matrix M1 exhibited the lowest (1.21 ± 0.40%; *p* = 0.014, F = 2.88, and DF_residuals_ = 310). In contrast, at the R5–R6 stage, the highest average parasitism rates were observed in matrix M2 (6.62 ± 1.21%) and M3 (6.49 ± 0.89%), while matrices M4 and M5 showed the lowest average parasitism rates (3.08 ± 0.82% and 3.28 ± 1.10%, respectively; *p* = 0.002, F = 3.85, and GL = 310). When the two phenological stages were compared, significant differences were found only in M1 (1.21 ± 0.39% and 5.67 ± 0.92% for Vn–R1 and R5–R6 respectively; *p* < 0.001, T = −4.42, and DF_residuals_ = 85) and M3 (2.42 ± 0.98% and 6.49 ± 0.89%, respectively; *p* = 0.003, T = −3.07, and DF_residuals_ = 123).

The wind speed during both releases was considered “Light breeze” by the Beaufort scale [22], with averages of 1.08 and 0.97 m∙s^−1^, respectively (Table 1 and Table 2). The mean temperature recorded at Vn–R1 was 23.86 °C, and 27.35 °C at R5–R6, with a relative humidity of 87.97% and 84.84% and an atmospheric pressure of 941.77 and 947.30 hPa, respectively (Table 2).

The semi-variograms were evaluated by verifying the isotropy of *T. podisi* parasitism during dispersal activity; that is, the direction did not affect the displacement and a single pattern described the spatial variability of the dispersal. The exponential model generated the best fit of r^2^ and, consequently, of the data, for the two generated semi-variograms (Figure 3). As anticipated, an inverse relationship between the parasitism rate and the distance from the point of release was observed (Figure 3). The maximum estimated dispersal range observed for *T. podisi* was 39.0 m in the Vn–R1 stage with a maximum parasitism rate of 42%, while in the R5–R6 stage the estimated maximum dispersal range was 30.9 m with a maximum parasitism of 73% (Figure 3).

We analyzed parasitism rates at three groups of distances (Figure 1). When the parasitoids were released in the Vn–R1 stage, there was a reduction in parasitism rate in the middle and outer boxes of about 3.43 and 4.26% compared with the inner box, respectively (*p* < 0.01; F = 7.65; DF_residuals_ = 315). When parasitoids were released in the R5–R6 stage, however, this difference among distances was not significant (*p* = 0.15; F = 1.88; DF_residuals_ = 315). Additionally, it was observed that the parasitism rate in the inner box was similar for both releases (*p* = 0.74, T = −0.32, and DF_residuals_ = 102.94), while the parasitism rate found in the middle and outer boxes was higher in R5–R6 compared to Vn–R1 (*p* > 0.01, T = −3.12, and DF_residuals_ = 150.14 and *p* < 0.01, T = −5.22, and DF_residuals_ = 344.05, respectively; Table 3).

## 4. Discussion

The semi-variogram and box analyses showed that the dispersal capacity of *T. podisi* may be affected by the phenological stages of soybeans. Although *T. podisi* exhibits higher mobility when the soybean is in the flowering stage (R1) compared to the pod-fill (R5–R6 stages), it was observed that the parasitism rate was slightly higher at the latest release date. This rate was equivalent across the three evaluated distances from the release site in the R5–R6 stage. *Euschistus heros* typically begin colonization during the late vegetative stage and the onset of the flowering stage (R1) and its densities continue to increase until the pod-maturation stage (R5) [2,23]. Detection of *E. heros* in each soybean field is critical to guide the timing of *T. podisi* release, while the different stages of the soybean might allow adjustments in release strategies. To achieve successful management through the mass release of egg parasitoids, it is extremely important that *E. heros* eggs are present at the time of *T. podisi* release [24]. Bueno et al. [2] recommend that *T. podisi* mass release be applied when the first stink bugs are found in soybean fields, which often occurs during the Vn–R1 soybean stage.

In addition to synchronizing with the host, the dispersal capacity of the parasitoid is a key factor for successful *E. heros* management. Previous studies have examined the *T. podisi* release method [2,12], but little is documented about its dispersal in soybean fields. Despite some differences between matrices in this study (Figure 2), our results indicate that *T. podisi* has a dispersal capacity of about 39.0 m in the Vn–R1 stage and 30.9 m at R5–R6. These findings align with a previous computational model that suggested a 25 m spacing between release points [25]. Thus, to ensure an effective distribution of *T. podisi* and maintain a well-protected coverage area, it is advisable to set release points at a maximum distance of 30 m from each other.

The difference observed in the calculated *T. podisi* dispersal capacity at different soybean stages could be influenced by a number of factors, including previous *T. podisi* release, plant architecture, size, and changing soybean plant chemistry. We found less average dispersal at the later soybean stage (R5–R6) but greater average parasitism at distances further from the release point (Figure 3, Table 2). Natural parasitism in the control plot did not increase between release dates; however, within experimental plots, infestation of natural eggs by the released parasitoids could have occurred and future research should quantify field establishment persistence.

Additionally, the plant architecture, size, and chemistry may also have impacted the dispersal capacity of *T. podisi*. Although we did not evaluate these parameters, it is likely that as the plants advance in phenological stage and their leaf area index increases [26], the maximum dispersal range of *T. podisi* is reduced in the R5–R6 stage compared to Vn–R1. More studies are required to test the effect of soybean plant architecture on parasitoid dispersal range, especially as influenced by wind speeds and direction. Volatile organic compounds released by soybean plants as they mature also alter parasitoid behaviors [27,28].

During the soybean reproductive stage, which coincides with the peak of natural infestation by stink bugs, the emission of volatiles tends to enhance the tritrophic interaction between plants, herbivores, and natural enemies. This leads to parasitoids exhibiting more dynamic behavior in the plant canopy, which could increase encounters with host eggs [29,30]. Host-fining and parasitism behaviors are also influenced by the chemical substances that are volatilized from fresh *E. heros* eggs. The use of cryopreserved eggs as sentinels in our experiments may have influenced the relatively low parasitism rates observed [31]. Although *E. heros* cryopreserved eggs are suitable for *T. podisi* laboratory mass rearing [8,9,32], the use of frozen eggs can slightly reduce the parasitism rate (%) by *T. podisi* [7]. Additionally, the use of poles with attached cryopreserved eggs did not present the same chemical compounds emitted by soybean plants during stink bug feeding prior to mating and oviposition [33].

It is also worth noting that we used *T. podisi* from a laboratory-reared population that has been maintained for more than 50 generations. The mating, foraging, and dispersal behaviors of these parasitoids may not be the same as those of populations found in nature. Future comparative studies with wild populations of *T. podisi* are required to detect the effects of the inbreeding experienced over generations in the laboratory. Individuals that are less fertile, less aggressive, and exhibit less flight activity have been observed previously in other species of Platigastridae [5].

The number of eggs offered (about 1890 eggs per plot), the time allowed for parasitism (24 h) and the use of cryopreserved eggs as sentinels might also explain the low parasitism rate observed. The number of eggs and distance might exceed *T. podisi*’s parasitism capacity in 24 h. Although in applied inundative biological control programs, the parasitism capacity holds greater significance than the lifespan of females in the field [8], *T. podisi* females might have needed more than 24 h for the pre-reproductive period to reach its full parasitism capacity in the field [34]. Bueno et al. [2] reported a 70% parasitism rate in soybean fields after *T. podisi* release; however, this was after three parasitoid releases and over a period of 27 days. Further studies are necessary to evaluate the reproductive potential and the parasitoid’s ability to locate hosts at different soybean stages under field conditions [7].

In the evaluation of the natural parasitism rate in our check areas, a very small percentage of parasitism was found in the study area. Under natural conditions, *T. podisi* is frequently found in *E. heros* eggs, reaching up to 100% parasitism rate, especially towards the end of the soybean cycle [35]. However, in our study, the natural parasitism level observed on both evaluation dates was as low as 0.32%. Therefore, although we used natural parasitism as a correction factor, it is likely that most of the parasitized eggs observed in the release areas originated from our *T. podisi* release. The reduced number of naturally occurring parasitoids in the area may reflect the intense use of broad-spectrum insecticides, including the organophosphate acephate during the previous soybean season (2019/2020).

Our observed parasitism rates in the field may also be underestimated by the methods we used. Cornelius et al. [36], found that only one in twelve parasitized eggs resulted in the emergence of a parasitoid when using sentinel eggs. Therefore, dissecting host eggs is necessary for accurately assessing the parasitism rate in the fields.

The dispersal capacity of parasitoids is often associated with wind speed and direction [37]. For instance, the parasitoid *Scelio fulgidis* (Hymenoptera: Scelionidae) can disperse up to 300 km daily on prevailing winds [24]. However, in our evaluations, the conditions were considered “Light breeze” during both releases [22], suggesting that further studies are needed to evaluate *T. podisi* dispersal under different weather conditions. Many studies of egg parasitoid dispersal indicate a decrease in parasitism rate with increasing distance from the release point [1,38,39,40]. As the parasitoid moves away from the release point, it has a greater opportunity to explore the area. This increased exploration leads to a higher energy demand and a greater exposure to biotic and abiotic factors, which contribute to natural mortality [1,39].

In developing any IPM program, it is important to consider life tables and all mortality factors of a pest. We found a maximum parasitism rate of 73% of sentinel eggs. Populations of *E. heros* are affected by diseases, other natural enemies [41,42,43,44] and by abiotic factors [45]. Barrufaldi et al. [34] documented 40% mortality just from molting in *E. heros* with the highest mortality occurring between the first and second-stage nymphs. At these stages, the stink bug does not compromise soybean yield or quality [46].

## 5. Conclusions

In conclusion, this work indicates that *T. podisi* has a dispersal capacity of between 39.0 and 30.9 m; thus, release points should be spaced at a maximum of 30 m apart to ensure field coverage. These results form the basis for logistics for the release of *T. podisi* in large areas and can be adapted to precision agriculture [14], optimizing the biological control program for application to large monoculture areas. Further studies are necessary to document the effects of weather conditions and other factors on *T. podisi*’s dispersal capacity and parasitism rates, and to align the spatial distribution dynamics with stink bug pests [23,47].

## Figures and Tables

**Figure 1 insects-15-00192-f001:**
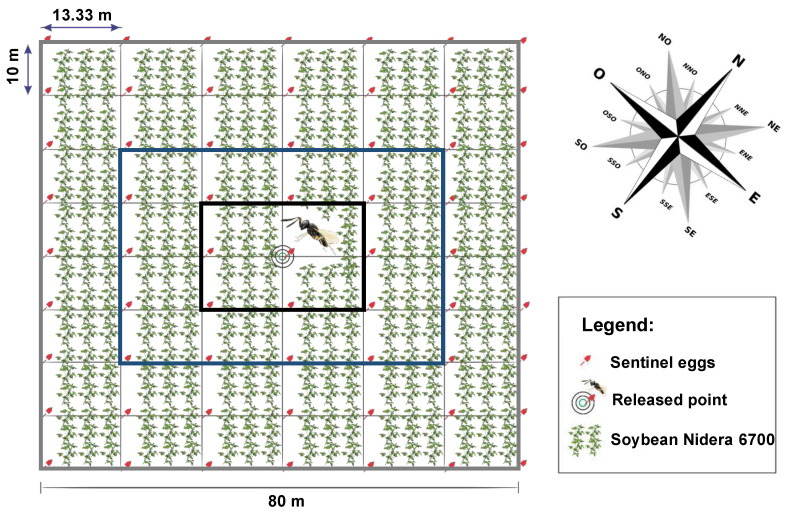
Experimental design one matrice of *Telenomus podisi* Ashmead (Hymenoptera: Platygastridae) on *Euschistus heros* eggs in a soybean field. A total of 63 equidistant points in 10 m vertically and 13.33 m horizontally in a square of 80 × 80 m. In the release centers of each of the six matrices demarcated in soybean cultivation (Nidera 6700 variety), 6500 females of *T. podisi* were released. Grey box = “Outer box”, blue box = “Middle box” and black box = “Inner box”.

**Figure 2 insects-15-00192-f002:**
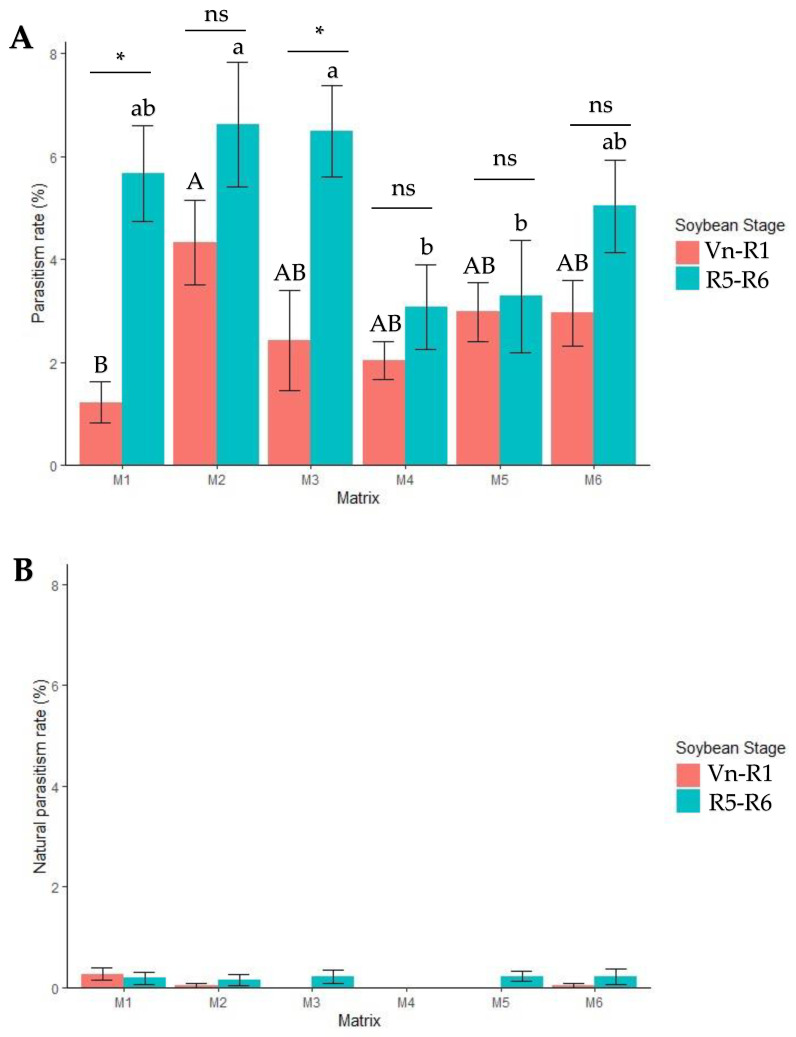
Parasitism of *Telenomus podisi* Ashmead (Hymenoptera: Platygastridae) on eggs of *Euschistus heros* artificially placed in 63 points in the field at stages Vn–R1 and R5–R6 in matrices 1 to 6 during the study of dispersibility in soybean. (**A**) = Parasitism rate (%) observed in *T. podisi* released plots, corrected with the natural parasitism. Bars (means ± standard error) followed by the same upper-case letter for soybean stage Vn–R1 and lower-case letter for soybean stage R5–R6 are not significantly different according to a Tukey test (α = 0.05); ns = no significant difference between R1 and R5 within the same matrix according to a *t*-test (α = 0.05); * = significant difference (*p* < 0.05) according to a *t*-test. (**B**) = Natural parasitism (%) in plots with no *T. podisi* release.

**Figure 3 insects-15-00192-f003:**
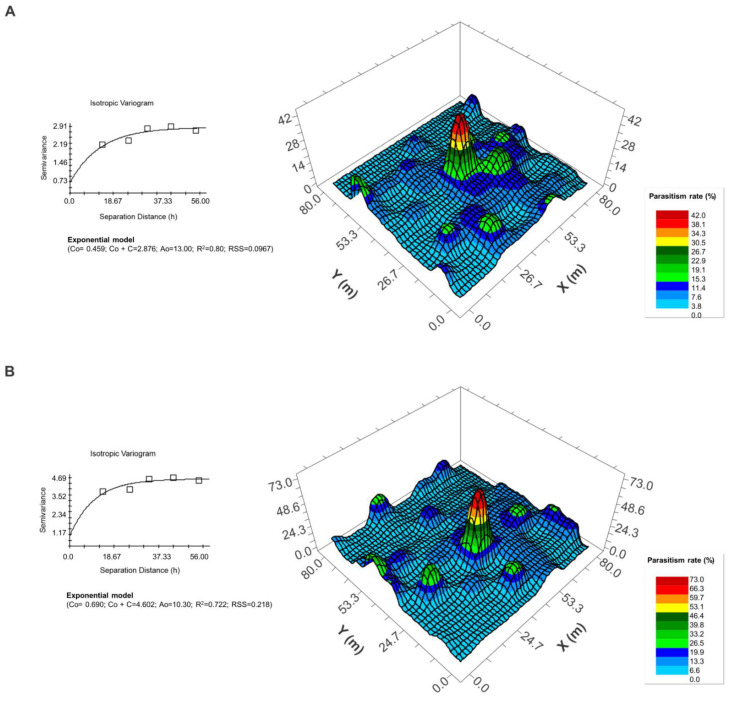
Isotropic semivariograms and point kriging maps generated from the percentages of parasitism of *Telenomus podisi* Ashmead (Hymenoptera: Platygastridae) on eggs of *Euschistus heros* artificially arranged at 63 points in the field at stages Vn–R1 (**A**) and R5–R6 (**B**) at Chiaparini farm, Santa Cruz do Rio Pardo, São Paulo, Brazil.

**Table 1 insects-15-00192-t001:** Meteorological elements at the release time of *Telenomus podisi* Ashmead (Hymenoptera: Platygastridae) in the field. Points collected from a digital and manual thermo-anemometer in the center of each release matrix (point 32).

Matrix	T °C	Wind (m/s)	Direction	RH %
Vn–R1				
M1	23.3	0.9	east	71
M2	22.3	1.0	east	68
M3	22.7	1.5	west	70
M4	22.1	1.5	east	71
M5	22.0	1.8	east	79
M6	22.5	2.1	east	83
R5–R6				
M1	23.3	1.5	west	77
M2	22.3	1.0	east	86
M3	22.7	2.0	west	72
M4	22.1	1.0	east	82
M5	22.0	1.0	east	59
M6	22.5	1.2	west	64

**Table 2 insects-15-00192-t002:** Mean meteorological elements during the 24 h of field dispersal tests of *Telenomus podisi* Ashmead (Hymenoptera: Platygastridae). Points collected from data generated by a meteorological station (Agroterenas S/A) on 6 January and 18 February 2020.

Stage	T °C	Wind (m/s)	RH (%)	PAtm (hPa)	Ppt mm	Date
Vn–R1	23.86	1.08	87.97	941.77	-	6 January 2020
R5–R6	27.35	0.97	84.84	947.30	0.64	18 February 2020

**Table 3 insects-15-00192-t003:** *Telenomus podisi* parasitism rate (%) ± standard error in the three different boxes schemes as illustrated in Figure 1. Analysis of variance conducted using the Box–Cox transformation method. Means ± SE followed by the same lowercase letter in a line are not significantly different using a Tukey test (α = 0.05), and uppercase letters in a column are not significant using a *t*-test (α = 0.05).

Stage	Parasitism Rate (%)		
	Inner Box	Middle Box	Outer Box
Vn–R1	6.10 ± 1.39 aA (*n* = 54)	2.67 ± 0.39 bA (*n* = 96)	1.84 ± 0.24 bA (*n* = 228)
R5–R6	6.81 ± 1.66 aA (*n* = 54)	5.16 ± 0.69 aB (*n* = 96)	4.56 ± 0.46 aB (*n* = 228)

## Data Availability

The data presented in this study are available on request from the corresponding author.

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
