# Peer review of "Optimizing the Release Pattern of Telenomus podisi for Effective Biological Control of Euschistus heros in Soybean"

_insects, 2024, doi:10.3390/insects15030192_

Round 1
Reviewer 1 Report
Comments and Suggestions for Authors
Manuscript is important and deserves being published but needs before major changes that I will highlight in the followings:
1) Text needs some small clarifications and improvements and some parts and more detailed information in others. Those suggestions are in the pdf file attached to this review.
2) My major concern with the manuscript is that I do not agree with conclusions and emphasis of the manuscript. Authors did an excellent and difficult job in the field but they did not prove soybean phenological stage impacted Telenomus podisi dispersion. Experimental designed is not appropriated for that.
They proved that there is variation along the plant development that is why a conservative approach of using lower distance should be advised during release. They mentioned that and this is perfect. However, the difference in dispersion is probably more due to plant size and plant distance among each other (the stand of plants) than due to plant phenological stage. Then, I do agree they emphasize plant phenological stage in the title and conclusion of the paper. Different plant cultivars, different grow habit (determinate or indeterminate), how many plants we have per meter will impact more dispersion than plant phenological stage. Then I suggest authors to change their perspective over the results and rewrite the manuscript that after this change and some other small editing in details and info that were not provided should be accepted because it is the type of papers that are need for Augmentative Biological Control. But I believe these change in perspective must be done before the manuscript be accepted for publication.

Comments on the Quality of English Languagesome editings will help
Author Response
Dear reviewer,
Thank you for your comments which improved our manuscript. Below, we highlight the changes, including a new title.
Reviewer #1’s comments:
Manuscript is important and deserves being published but needs before major changes that I will highlight in the followings:
1) Text needs some small clarifications and improvements and some parts and more detailed information in others. Those suggestions are in the pdf file attached to this review.
All the suggestions were accepted and incorporated into the manuscript using tracking mode.
2) My major concern with the manuscript is that I do not agree with conclusions and emphasis of the manuscript. Authors did an excellent and difficult job in the field but they did not prove soybean phenological stage impacted Telenomus podisi dispersion. Experimental designed is not appropriated for that.
Reviewer one makes an excellent point and we have addressed this limitation in our study in the discussion. We also made changes in the conclusion and removed the emphasis of soybean phenological stage.
We have changed the title: Optimizing release pattern of Telenomus podisi for effective biological control of Euschistus heros in soybean
Also, we added the paragraph L267-272: “Additionally, the plant architecture, size, and chemistry may also have impacted the dispersal capacity of T. podisi. Although we did not evaluate these parameters, it is likely that as the plants advance in phenological stage and their leaf area index increases [26], the maximum dispersal range of T. podisi is reduced in the R5-R6 stage compared to Vn-R1. More studies are required to test the effect of soybean plant architecture on parasitoid dispersal range, especially as influenced by wind speeds and direction.”
3) They proved that there is variation along the plant development that is why a conservative approach of using lower distance should be advised during release. They mentioned that and this is perfect. However, the difference in dispersion is probably more due to plant size and plant distance among each other (the stand of plants) than due to plant phenological stage. Then, I do agree they emphasize plant phenological stage in the title and conclusion of the paper. Different plant cultivars, different grow habit (determinate or indeterminate), how many plants we have per meter will impact more dispersion than plant phenological stage. Then I suggest authors to change their perspective over the results and rewrite the manuscript that after this change and some other small editing in details and info that were not provided should be accepted because it is the type of papers that are need for Augmentative Biological Control. But I believe these change in perspective must be done before the manuscript be accepted for publication.
Thank you for all the valuable suggestions and insights. We appreciated and accepted all of them. We have changed the perspective as we edited the text and the title of our manuscript in order not to focus on the soybean phenological stage as one of the main sources of variation in the T. podisi dispersal range and its impacts on biological control releases. Thank you!
Reviewer 2 Report
Comments and Suggestions for Authors
The manuscript presents data of interest to biological control practitioners, and the results are useful in decision making. However, a little more detail is required in the methodology. Suggestions are included in the attached file

Author Response
Dear Reviewer 2,
Thank you for your comments. We added details to the methodology and adjusted the title and discussion focus away from phenological stage based on comments from reviewer1.
The manuscript presents data of interest to biological control practitioners, and the results are useful in decision making. However, a little more detail is required in the methodology. Suggestions are included in the attached file.
Thank you for all the valuable comments and suggestions. All of them were addressed, and the information was added or edited in the text.
Round 2
Reviewer 1 Report
Comments and Suggestions for Authors
Authors edited the manuscript accordingly to the suggestions and now I believe it can be accepted for publication. I do not feel qualified enough to check English neither journal´s rules and formatting.